# Reconfigurable Continuous Meta-Grating for Broadband Polarization Conversion and Perfect Absorption

**DOI:** 10.3390/ma14092212

**Published:** 2021-04-26

**Authors:** Yijia Huang, Tianxiao Xiao, Zhengwei Xie, Jie Zheng, Yarong Su, Weidong Chen, Ke Liu, Mingjun Tang, Ling Li

**Affiliations:** 1College of Physics and Electronic Engineering, Sichuan Normal University, Chengdu 610101, China; zzwxie@aliyun.com (Z.X.); zhenghe_880904@126.com (J.Z.); suyarong@126.com (Y.S.); cwd_ck@163.com (W.C.); lkworld@uestc.edu.cn (K.L.); tmj200000808@aliyun.com (M.T.); 2Physik-Department, Lehrstuhl für Funktionelle Materialien, Technische Universität München, James-Franck-Straße 1, 85748 Garching, Germany; tianxiao.xiao@ph.tum.de

**Keywords:** metasurface, continuous structure, phase change material, polarization conversion, electromagnetic absorption

## Abstract

As promising building blocks for functional materials and devices, metasurfaces have gained widespread attention in recent years due to their unique electromagnetic (EM) properties, as well as subwavelength footprints. However, current designs based on discrete unit cells often suffer from low working efficiencies, narrow operation bandwidths, and fixed EM functionalities. Here, by employing the superior performance of a continuous metasurface, combined with the reconfigurable properties of a phase change material (PCM), a dual-functional meta-grating is proposed in the infrared region, which can achieve a broadband polarization conversion of over 90% when the PCM is in an amorphous state, and a perfect EM absorption larger than 91% when the PCM changes to a crystalline state. Moreover, by arranging the meta-grating to form a quasi-continuous metasurface, subsequent simulations indicated that the designed device exhibited an ultralow specular reflectivity below 10% and a tunable thermal emissivity from 14.5% to 91%. It is believed that the proposed devices with reconfigurable EM responses have great potential in the field of emissivity control and infrared camouflage.

## 1. Introduction

Recently, metasurfaces, the planar version of metamaterials, have attracted widespread attention, owing to their large degree of freedom for manipulating electromagnetic (EM) waves at subwavelength scale [1,2,3,4]. By introducing localized interactions between meta atoms and EM waves, full vectorial control of light can be achieved, in terms of amplitude, phase, and polarization, which enables many fascinating applications, including anomalous beam deflection [5], sub-diffraction imaging [6,7], perfect EM absorption [8,9,10], and meta-holography [11,12,13], to name but a few. Since metasurfaces possess exotic properties that natural materials do not have, they can also realize counterintuitive functionalities such as carpet cloaking [14,15] and daytime radiative cooling [16,17]. These landmark achievements for EM wave engineering have laid the foundation for the upcoming Engineering Optics 2.0. [18,19]

Although great progress has been made in the field of metasurfaces, traditional metasurface devices are often limited by low working efficiencies and narrow operation bandwidths, due to the resonant nature of discrete meta atoms, which hinder their practical applications [20,21]. In order to further enhance the performance of metasurfaces, a potential solution is to employ continuous or quasi-continuous meta atoms to replace their discrete counterparts. Therefore, a subwavelength version of traditional diffractive grating, called meta-grating, has been proposed to further improve the working bandwidth and efficiency. Previously reported works have demonstrated that by properly designing meta-gratings based on different physical mechanisms, highly efficient broadband manipulation of EM waves can be achieved [22,23,24,25]. Furthermore, it has also been proved that more functionalities can be realized by moving meta-gratings towards their streamlined counterparts [26]. For example, Huang et al. achieved a perfect broadband beam deflection using a continuous metasurface composed of trapezoidal antennas [27]. Yuan et al. proposed a kind of nanoarc metasurface that can achieve broadband high-efficiency beam splitting and meta-holograms [28]. Luo et al. established the theory of catenary electromagnetics, which linked the catenary functions to the inner physics of metasurfaces and explicitly explained the advantages of continuous meta-gratings [26]. In addition, his group has designed a variety of high performance continuous or quasi-continuous metasurface devices that can achieve vortex beam generation [29,30], airy beam generation [20,31], coherent absorption [32], sub-diffractive imaging [33], and wide field-of-view imaging [34]. Although the above mentioned works further improved the performance of metasurfaces, how to integrate more functionalities and achieve reconfigurable devices is still challenging.

Here, by combining a high performance meta-grating with a tunable phase change material (PCM), Ge_2_Sb_2_Te_5_ alloy (GST-225, simply GST hereafter), a reconfigurable device with a simple geometry is proposed, which works in the infrared region. To further illustrate this issue, Figure 1 depicts a schematic of the proposed device. It can be inferred from Figure 1a that the device can achieve polarization conversion for both linear and circular incidences when the GST is in amorphous state (A-state). In this case, the right-handed polarized (RCP) and transverse electric (TE) polarized incidence can be efficiently converted to their left-handed polarized (LCP) and transverse magnetic (TM) counterparts, respectively, and vice versa. When tuning the crystallization level to a crystalline state (C-state) by outer stimuli, the device will behave as a perfect absorber, regardless of the incident polarizations, as shown in Figure 1b. It should be mentioned that although previously reported works based on PCM have achieved reconfigurable devices with various functionalities, they could only modulate one aspect of the EM wave, such as changing the absorption peak or altering the implemented phase [35,36,37,38,39,40,41,42]. Whereas, the presented device can manipulate the polarization and amplitude in two crystallization levels, which may enable more fascinating applications in optics and photonics.

## 2. Design Principles and Simulation Results

Figure 2a depicts a conceptual illustration of the unit cell, composed of a silicon (Si) grating, GST spacer, and Au reflective layer. Corresponding geometrical parameters are given in the figure caption. Considering the symmetry of the proposed device, a finite element method in CST Microwave Studio was employed to obtain the simulated results, with unit cell boundaries in the *xy* directions, and the open boundary in the *z* direction. In the simulation process, the permittivities of the GST were adopted from measured results in a previous work [37]. The real part and imaginal part of the permittivities of amorphous and crystalline GST within 26 to 32 THz are shown in Figure 2b,c. It can be inferred that when the GST is in an A-state it is an ideal dielectric material with no loss, while it behaves as a semiconductor with considerable loss when it changes to a C-state. The permittivity of Si was obtained from Palik [43], and the permittivity of Au was described by Drude’s model [36].
(1)$$\varepsilon_{Au}=\varepsilon_{\infty }-\frac{\omega_{p}^{2}}{\omega (\omega +i\gamma )}$$
where the plasma frequency *ω_p_* and collision frequency *γ* are chosen to be 1.32 × 10^16^ rad/s and 131.8 THz, respectively. *ε*_∞_ is the relative permittivity when the frequency is infinite and is set as 9.1. Figure 2d shows the simulated results under circularly incidence when the GST is in an A-state. The reflected amplitude of the cross-polarized component above 90% ranges from 26 to 29.2 THz. Since the Au reflectivity is thick enough to prevent transmission, the absorption *A* can be calculated by
(2)$$A=1-r_{co}^{2}-r_{cr}^{2}$$
where *r_co_* and *r_cr_* are the co- and cross-polarized amplitudes, respectively. Thus, the corresponding absorption is shown in Figure 2e, which is below 14.5% from 26 to 29.2 THz. It should be mentioned that although the results were obtained under a circularly polarized incidence, the proposed device can achieve the same polarization conversion performance under a linearly polarized incidence when the azimuthal angle between the electric vector of the incidence wave and the main axis of the meta-grating is 45°, as shown in Figure 1a [36]. Therefore, the presented meta-grating in the A-state can be treated as a highly-efficient broadband polarization convertor for both circular and linear polarizations. The simulated results when the crystallization level of the GST changed to the C-state are shown in Figure 2f,g. It can be inferred that the reflected amplitudes for both co- and cross-polarized components were less than 30% from 28.1 to 32 THz, and the corresponding absorption was larger than 91%. In this case, the device behaves as a perfect broadband absorber, with ultra-low reflection. Therefore, the above mentioned meta-grating can perform as a dual-mode reconfigurable device, with two distinct functionalities, in terms of polarization and amplitude control.

To explore the physical origin of the proposed device at different crystallization levels, Figure 3 depicts the simulated results under a linearly polarized incidence at 29 THz. The directions of the electric vector for the TE and TM incidence are labelled in the insets of Figure 3a,b, respectively. Obviously, a strong localized resonance can be observed as shown in Figure 3a under the TE incidence, while the interaction between the meta-grating and incident TM wave in Figure 3b is much weaker than that in Figure 3a. Therefore, due to the resonant differences in these two cases, relative phase differences, ∆P, between TE and TM polarizations can be obtained. To further demonstrate this issue, Figure 3c shows the simulated phase responses. It can be inferred that ∆P is about π from 26 to 29.2 THz (the gray region depicts ∆P = π ± 0.1π). Thus, the corresponding polarization conversion efficiency is near unity in this case. Moreover, in order to shed light onto the absorption mechanism of the proposed meta-grating when the GST is in the C-state, simulated energy flows of magnetic and electric fields under TE and TM polarized incidence are shown in Figure 3d,e, respectively. The circumfluence of the magnetic field in Figure 3d demonstrates the existence of multiple electric dipoles, and the current loop of the electric field in Figure 3e indicates the existence of a magnetic dipole. Therefore, the absorption mechanism for the TE and TM incidence can be explained by electric dipole resonance and magnetic dipole resonance, respectively. The simulated absorptions for both cases are shown in Figure 3f, and the absorption bandwidth under a TM incidence is larger than that under a TE incidence, which demonstrates the typical features of magnetic (broadband) and electric (narrowband) resonances [21].

A potential application for the proposed meta-grating is to achieve an ultralow specular reflection and tunable emissivity in broad wavebands, as shown in Figure 4a. As such, a quasi-continuous device was designed with a chessboard-like configuration, consisting of orthogonally arranged meta-gratings [44]. When the GST is in A-state, according to the theory of geometric phase (also known as Pancharatnam–Berry phase) [45], the proposed meta-grating can be treated as a highly efficient phase retarder, since the cross-polarized amplitude is much larger than its co-polarized counterpart. In this case, the reflected phase for a crossed polarized wave is twice the orientation angle of the meta-grating. Therefore, the reflected beam will be guided in well defined directions, with low specular reflection, due to the orthogonally arranged adjacent pixels. According to Kirchhoff’s law of thermal radiation that the directional spectral emittance is equal to the directional spectral absorption [46], the emissivity in this case is lower than 14.5%, as shown in Figure 2e. When the GST is in C-state, since the meta-grating can effectively absorb the EM waves, the proposed symmetric chess-board structure can achieve a polarization independent absorption, with ultralow reflection. Differently from the former A-state performance, the emissivity can reach up to 91% in this case. Therefore, the proposed quasi-continuous device can serve as an infrared invisible metasurface, with tunable thermal emission properties. To further demonstrate this, subsequent full-wave simulations were performed to validate the performance of the designed metasurface, and the simulated results are shown in Figure 4b–e. The time domain solver of CST Microwave Studio was employed for the simulations, with periodic boundaries in the *xy* directions and open boundary in the *z* direction. To simplify the case and without a loss of generality, *n* was chosen as *n* = 5 for the simulation. It can be inferred from Figure 4b that the reflected wave is redirected in four diagonal directions, and the reflection along the specular direction can hardly be observed. Figure 4c depicts the farfield scattering patterns when the GST is in C-state, and ultralow reflection in all directions can be observed due to the highly absorptive nature of the meta-gratings. In fact, since the linearly polarized incidence can be decomposed into two circularly polarized incidences with opposite handedness, but the same amplitude, similar performances as in Figure 4b,c can be achieved for arbitrary polarization incidences, due to the fourfold geometrical symmetry of the device. For comparison, Figure 4d shows corresponding results of an unpatterned gold plate, with the same dimensions as the proposed metasurface. It is obvious that the specular reflectivities for the metasurface in both A- and C-states are much smaller than for the metallic plate. To further demonstrate the broadband performance of the proposed quasi-continuous device, Figure 4e shows the simulated specular reflectivity for the above-mentioned three cases, from 26 to 32 THz. It can be inferred that the specular reflectivity for the metallic plate is near unity in broadband, while a significant reduction of reflection for the metasurface can be observed. When the GST is in A-state, the average specular reflectivity below 10% ranges from 26 to 28.3 THz, and the emissivity of the device is below 14.5%. Furthermore, when the GST is in C-state, the average specular reflectivity below 10% ranges from 26 to 32 THz, and the emissivity of the device is above 91%. It can be observed that the operation bandwidths of the quasi-continuous metasurface in A-state (26 to 29.2 THz) and C-state (28.1 to 32 THz) slightly deviate from the performance of the meta-grating (26 to 28.3 THz for A-state, 26 to 32 THz for C-state) in Figure 2, which can be explained by the mutual coupling between adjacent orthogonal pixels, which further enhanced the absorption, but reduced the polarization conversion ratio (PCR).

To further demonstrate the superior performance of the continuous unit cell, Figure 5 depicts the design and performance with different geometric parameters for comparison. As shown in Figure 5a, the geometric parameters such as P, *w,* and the thicknesses of the materials are the same as those in Figure 2a. Figure 5b shows the simulated reflected amplitudes under circularly polarized incidence when *l* changes from 2.8 to 5.8 μm, with 1 μm as an interval. It can be inferred that the cross polarized components, when *l* = 3.8 μm, are much smaller than the other three cases, due to the fact that such a unit cell is isotropic with *l* ≈ *w*. Additionally, when *l* = 5.8 μm (*l* = P), the proposed unit cell can achieve higher polarization conversion efficiency and wider operation bandwidth than in other cases. In this case, the unit cell with *l* = P is the same as that in Figure 2a with a continuous profile. To further demonstrate this issue, Figure 5c shows the frequency dependent PCR for the above four cases. Obviously, the performance with *l* = 5.8 μm is better than the other cases, with a higher PCR value and broader bandwidth. It should be mentioned that although the PCR value for *l* = 4.8 μm is comparable to that with *l* = 5.8 μm, the absorption in the former case is much larger. The average absorption for *l* = 4.8 μm is larger than 22.7%, and the corresponding absorption is less than 14.5% for *l* = 5.8 μm. Therefore, the above analysis further demonstrates that the proposed continuous unit cell possesses a higher working efficiency, wider operation bandwidth, and lower Ohmic loss than its discrete counterpart.

## 3. Conclusions

Except for the aforementioned application in Figure 4, the proposed meta-grating can be designed to achieve arbitrary phase engineering, such as beam deflection, focusing, and holography in A-state, based on geometric phase by simply rotating the meta-gratings to meet the required phase distributions. While broadband perfect absorption can still be realized when the GST changes to C-state. Moreover, although subsequent experiments were not performed in the presented work, many related works have proven that the fabrication of the designed metasurfaces can be implemented with a low cost and high precision [47,48]. Previous works also demonstrated that the crystallization and re-amorphization of GST can be successfully achieved by several methods [49,50,51], such as thermal annealing, electrical stimulus, and laser pulse illumination. Furthermore, inspired by previous streamlined surfaces, the proposed meta-grating can be treated as a potential supplement in the field of catenary electromagnetics.

In summary, we have proposed a simple yet powerful design methodology for reconfigurable meta-gratings in the infrared region. By combining the superior performance of a continuous structure with tunable PCMs, the designed broadband dual-mode device has the capability of achieving efficient polarization conversion, with an average efficiency over 90% and perfect EM absorption over 91%, when the GST is at different crystallization levels. Moreover, the performance comparison between the proposed unit cell and its discrete counterparts further proved that the continuous meta-grating exhibits a higher working efficiency, wider operation bandwidth, and lower Ohmic loss. We believe that the proposed reconfigurable device with a simple geometry has great potential for practical applications in beam steering, encrypted information storage, and thermal emission control.

## Figures and Tables

**Figure 1 materials-14-02212-f001:**
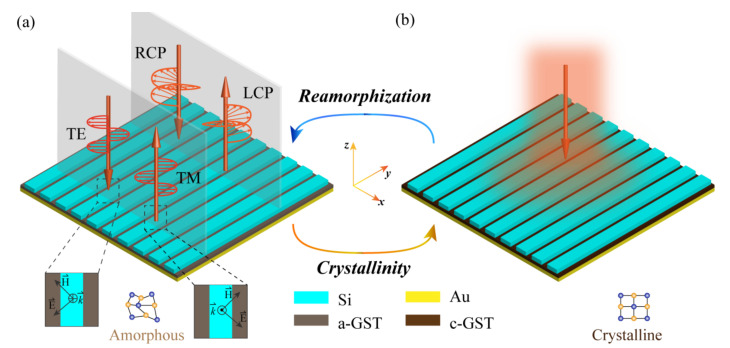
The schematic diagram of the proposed meta-grating. It can achieve broadband polarization conversion when the PCM is in amorphous state (**a**), and perfect EM absorption when the PCM changes to a crystalline state (**b**). The bottom insets in (**a**) depict the azimuthal angle between the electric vector of the EM wave and the main axis of the meta-grating. The crystallization levels of GST in (**a**,**b**) do not represent the real molecular structures.

**Figure 2 materials-14-02212-f002:**
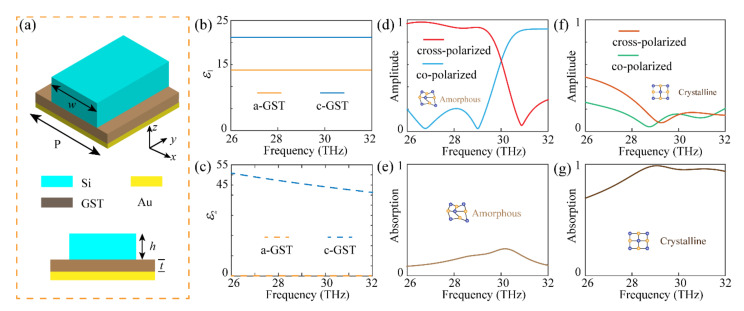
The schematic diagram of the unit cell and simulated performance of the device. (**a**) The three dimensional (top) and section view (bottom) of the unit cell. The corresponding geometrical parameters are P = 5.8 μm, *w* = 3.77 μm (filling ratio 0.65), *h* = 2 μm, and *t* = 0.3 μm. (**b**,**c**) Measured real part of permittivities *ε*_1_ (**b**), and imaginal part *ε*_2_ (**c**) of the amorphous and crystalline GST (a-GST and c-GST). (**d**,**e**) Reflected amplitude of the cross- and co-polarized components under circularly polarized incidence (**d**), and the corresponding absorption (**e**) when the GST is in A-state. (**f**,**g**) Reflected amplitude of the cross- and co-polarized components under circularly polarized incidence (**f**), and the corresponding absorption (**g**) when the GST is in C-state.

**Figure 3 materials-14-02212-f003:**
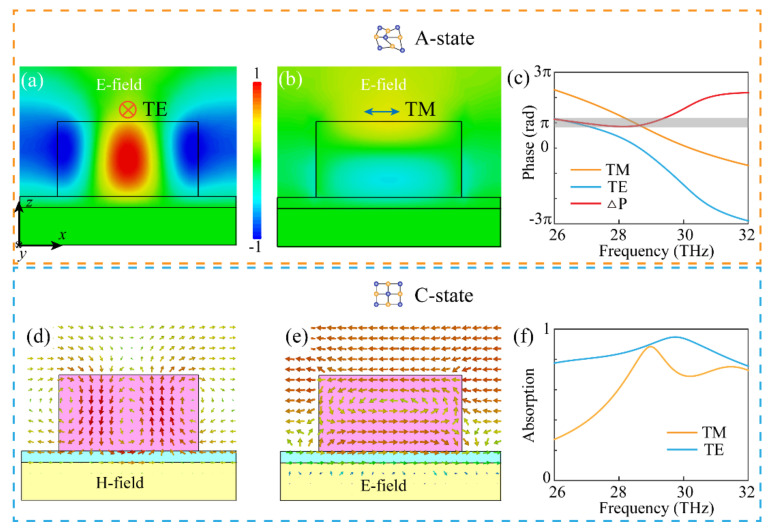
Simulated results under a linearly polarized incidence at 29 THz. (**a**,**b**) Simulated normalized electric field distributions under TE and TM polarized incidence in A-state. (**c**) Relative phase differences between TE and TM polarizations. (**d**,**e**) Simulated energy flows of magnetic (**d**), and electric (**e**) fields under a TE and TM polarized incidence in C-state. (**f**) Absorption performance for TE and TM incidence. The insets in (**a**,**b**) define the polarization direction of the TE and TM incidences.

**Figure 4 materials-14-02212-f004:**
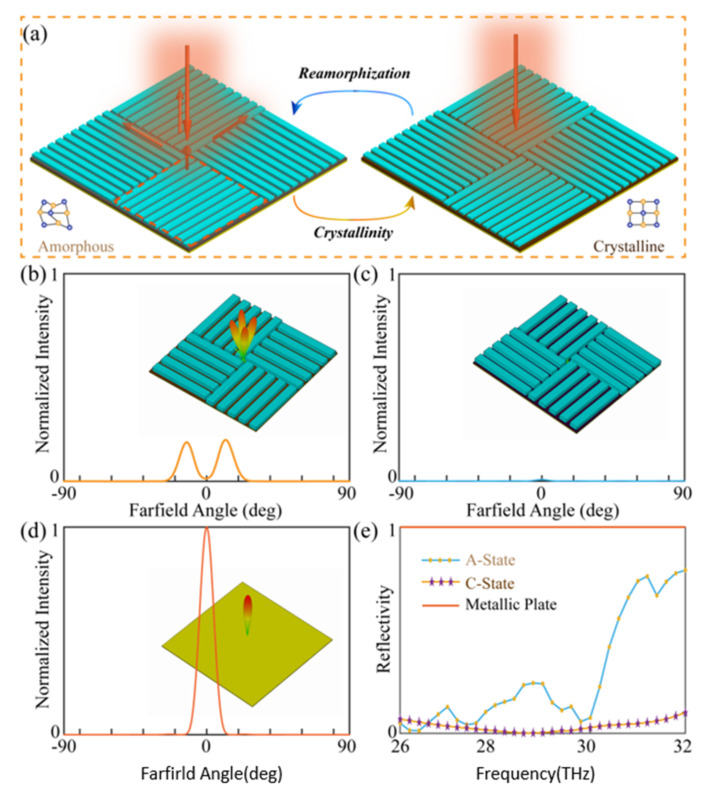
The performance of the quasi-continuous meta-grating. (**a**) Schematic diagram of the chessboard-like meta-grating. Each pixel marked with dotted lines consists of *n* strips of the meta-grating. (**b**) Farfield scattering patterns when the GST is in A-state at 26.2 THz. (**c**) Farfield scattering patterns when the GST is in C-state at 29 THz. (**d**) Farfield scattering patterns for a metallic plate at 29 THz. The insets in (**b**,**d**) are the corresponding three-dimensional scattering patterns. (**e**) Simulated specular reflectivities of the quasi-continuous meta-grating and the metallic plate for a linear polarized incidence.

**Figure 5 materials-14-02212-f005:**
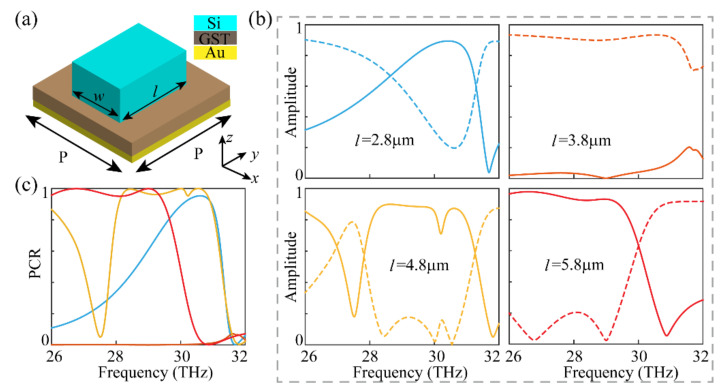
The design and performance with different geometric parameters. (**a**) The schematic diagram of the unit cell. P, *w,* and the thicknesses of the materials are the same as those in Figure 2a. (**b**) Simulated reflected amplitudes with different values of *l*. The solid and dotted lines are the cross- and co-polarized results respectively. (**c**) Corresponding PCR with different values of *l*.

## Data Availability

The data that support the findings of this study are available within the article.

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
