# Peer review of "Reconfigurable Continuous Meta-Grating for Broadband Polarization Conversion and Perfect Absorption"

_materials, 2021, doi:10.3390/ma14092212_

Round 1

Reviewer 1 Report

In this work presented by Y. Huang et. al. is proposed a metasurface design for a Broadband Polarization Converter which can be switched on a Perfect Absorber. I cannot recommend this work for publication as it is not novel and cannot add any useful insight on the readers of this journal. Realisation of such device is not feasible, as there is not capping layer on the design of the device.

Here are some works with similar designs & results:

Related to absorbers with phase change materials:

Cao, T., Wei, Cw., Simpson, R. et al. Broadband Polarization-Independent Perfect Absorber Using a Phase-Change Metamaterial at Visible Frequencies. Sci Rep 4, 3955 (2014).

Guo, Z., Yang, X., Shen, F. et al. Active-Tuning and Polarization-Independent Absorber and Sensor in the Infrared Region Based on the Phase Change Material of Ge2Sb2Te5 (GST). Sci Rep 8, 12433 (2018). 

Also to phase change polarisation converters:

Cao, T., Wei, C. & Mao, L. Numerical study of achiral phase-change metamaterials for ultrafast tuning of giant circular conversion dichroism. Sci Rep 5, 14666 (2015).

Xiao, Z., Zou, H., Zheng, X. et al. A tunable reflective polarization converter based on hybrid metamaterial. Opt Quant Electron 49, 401 (2017).

Reviewer 2 Report

Dear Editor, thank you for the trust and the opportunity to provide a review of the work " Reconfigurable Continuous Meta-grating for Broadband Polarization Conversion and Perfect Absorption " by Yijia Huang, Tianxiao Xiao, Zhengwei Xie, Jie Zheng, Yarong Su, Weidong Chen, Ke Liu, Mingjun Tang and Ling Li.

 The paper is dedicated to the study of the superior performance of continuous metasurface combined with reconfigurable property of phase change material (PCM). A dual-functional meta-grating is proposed in infrared region that can achieve broadband polarization conversion over 90% when PCM is in amorphous state and perfect EM absorption larger than 91% when PCM changes to crystalline state.

 The manuscript is well structured and very well written and I consider it is suitable for publication after minor revision.

I have the following suggestions and comments:

Comment 1: The refractive indexes of the GST and the Si were obtained from experiment and Palik data respectively, while the refractive index of gold was modeled using a Drude model. As far as I know, the Drude model agrees well with the experimental data for silver, while this agreement is worse for gold. Why wasn't the data taken from Johnson and Christy paper? I believe that the dispersion of gold can significantly change the obtained results. The authors should clarify this point.

Comment 2: The field in Figure 3a is similar to the field at the bound state in the continuum wavelength, since it is localized in the lattice plane. How can this be explained?

Comment 3: The legend should be added in fig 5a.

Reviewer 3 Report

- Sectioning of the paper should be rearranged because in the present form the section 3.Discussion does not separate the discussion from the presentation of the results, the latter is just continuing in this section.

- The metallic grating in “…In this case, the reflected phase for crossed polarized wave is twice the orientation angle of the metallic grating…” on Page 5 should be clarified. In the text authors use meta-grating.

- “…It can be inferred that the cross polarized components when l=3.8 μm is much smaller than other three cases due to the fact that such unit cell is isotropic with 217 lw...” on Page 7, contradicts Figure 5b, where cross-polarized component for l=3.8 mkm is higher than for 2.8, 4.8, and 5.8 mkm.

- Conclusions are badly written. For example, there are non-conclusive sentences included, like “…Subsequent numerical simulations were also performed to reveal the corresponding inner physics…” or sentences providing no conclusive message, just a general statement, like “…Besides, the performance comparison between continuous unit cell with its discrete counterparts further proved the advantages of the presented structure with higher working efficiency, wider operation bandwidth and lower Ohmic loss…”. What is meant with the “presented structure”, because structures of continuous, quasi-continuous and of different design were investigated?

- The x, y, z-axis should be located also in Figure 3.

Reviewer 4 Report

This paper mainly presents reconfigurable continuous meta-grating for broadband polarization conversion and perfect absorption. This makes the topic very interesting for optics and photonics.

The introduction provides a good, generalized background of the topic that quickly gives the reader an appreciation of engineering optics.

I think the motivations presented in the paper are clear and not in dispute.

The paper fully describes the posed question.

Topic presented in this paper concentrates on a reconfigurable continuous meta-grating. Thus, authors decided to show their own design of a meta-grating. The presented device gives possibility to manipulate the polarization and amplitude in two crystallization levels respectively, which may enable  more fascinating applications in optics and photonics (particularly in e.g. beam steering, encrypted information storage, and thermal emission control).

The experimental apparatus is quite modern and very suitable for this study.

The literature cited  in this article is relevant to the study.

This paper is very well written. The text for reader is very clear and easy to read.

The conclusions of  the paper are consistent with the evidence and argument presented. These conclusions and the discussion undertaken address  the main question posed.

The work is done at a very good science level and can be recommended for publication.

Round 2

Reviewer 1 Report

I am still not convinced on the impact and novelty of this work, so I cannot recommend for publication in this journal.

-The authors in their response add some simulation, however they should take into account that the protective layer should exist above and below the GST film, as the metallic/GST interfaces are very sensitive to the switching process, therefore the results presented are not convincing either. see below ref:

Gholipour, B., Zhang, J., MacDonald, K.F., Hewak, D.W. and Zheludev, N.I. (2013), An All‐Optical, Non‐volatile, Bidirectional, Phase‐Change Meta‐Switch. Adv. Mater., 25: 3050-3054. https://doi.org/10.1002/adma.201300588

-The references authors quote are related to experimental works with thermal bias, since your submission is related to solely simulation results, they have to be more thorough related to the reversible switching of the proposed devices.